- Changes in carbon stocks of Fagus forest ecosystems along 1
- an altitudinal gradient on Mt. Fanjingshan in Southwest 2
- China 3
- Qiong Cai<sup>1</sup>, Chengjun Ji<sup>1</sup>,\*, Xuli Zhou<sup>1</sup>, Wenjing Fang<sup>1</sup>, Tianli Zheng<sup>1</sup>, Jiangling Zhu<sup>1</sup>, 4
- Lei Shi<sup>2</sup>, Haibo Li<sup>2</sup>, Jianxiao Zhu<sup>1</sup> and Jingyun Fang<sup>1</sup> 5
- 6
- <sup>1</sup> Department of Ecology, College of Urban and Environmental Sciences, and Key Laboratory for Earth
- 7 Surface Processes of the Ministry of Education, Peking University, Beijing 100871, China
- 8 9 <sup>2</sup> Administration Bureau of Guizhou Fanjingshan National Nature Reserve, Jiangkou 554400, Guizhou, 10 China
- Correspondence to: Chengjun Ji (jicj@pku.edu.cn) 11
- 12 Tel: +86-10-6276 5578, Fax: +86-10-6275 6560

Page 2

13 Abstract: There are four components of carbon (C) pools in a natural forest ecosystem: vegetation, soil, litter 14 and woody debris. Quantifying these C pools and their contributions to forest ecosystems is important in 15 understanding C cycling in forests. Here, we investigated these four C pools in nine beech (Fagus L., Fagaceae) 16 forests along an altitudinal gradient in southwest China. We found that the C pools of beech forest ecosystems 17 ranged from 190.7 to 503.9 Mg C ha<sup>-1</sup>, mainly attributed to vegetation C (accounting for 33.7–73.9%) and 18 soil C (accounting for 24.6-65.4%). No more than 4% of ecosystem C pools were stored in woody debris 19 (0.25-3.4%) and litter (0.2-0.7%). Ecosystem C storage increased significantly with altitude, where the 20 vegetation and woody debris C pools increased concomitantly with increasing altitude, while those of litter 21 and soil exhibited no significant variations. The forest stand age was found to be a key driver of such altitudinal 22 patterns, especially for vegetation C storage. The present study provides reliable data for understanding the 23 structure and function of Chinese beech forests, and emphasizes the importance of considering the influence 24 of stand age on C accumulation.

Keywords: ecosystem carbon storage, carbon components, Fagus forests, stand age, altitudinal gradient

Page 3

## 27 1 Introduction

Forests are among the most vital carbon (C) pools on earth, where these C pools play a key role in C cycling in 29 terrestrial ecosystems (Pan et al., 2011). There are four components of C stocks in a natural forest ecosystem: 30 vegetation, woody debris, litter, and soil (IPCC, 2013). The quantities and contributions of these four 31 components of forest C stocks are affected by numerous factors, such as climate (Aplet and Vitousek, 1994); 32 stand age (Pregitzer and Euskirchen, 2004); stand conditioning; the origin, type, and structure of the forest (Niu 33 et al., 2009); and even management (He et al., 2013) and disturbance (Zhang and Wang, 2010), where climate 34 and stand age (or time since disturbance) are among the key drivers (Aplet and Vitousek, 1994; Gower et al., 35 1997). It has been reported that the C pools in the forest ecosystems of China were dominantly attributed to 36 those of vegetation and soil (Fang et al., 2007; 2014). In addition, the C storage and ratios of litter and woody 37 debris in China's forests were far lower than those of other temperate forests around the globe (Zhu et al., 38 2017a). Recently, ecologists have not only focused on the C storage attributed to vegetation, but have also given 39 more attention to C storage in the soil (Yang et al., 2008, 2014; Chen et al., 2015), as well as to that of litter and 40 woody debris (Hu et al., 2015; Zhu et al., 2017b). To shed more light on how forest ecosystems respond to the 41 ongoing climate change and human disturbance, it is essential to evaluate the contributions of all aspects of C 42 storage in forest ecosystems in a comprehensive manner.

With increases in altitude, there forms an environmental gradient which mirrors the complicated variations 44 in climatic factors (e.g. air temperature and precipitation), topography, and other environmental conditions 45 (Körner, 2007). The C storage of vegetation generally tended to decrease over long altitudinal gradients due to 46 climatic and nutrient limitations on growth (Vitousek et al., 1992; Aplet and Vitousek, 1994; Leuschner et al., 47 2007; Alves et al., 2010; Zhu et al., 2010), while shorter altitudinal gradients might exhibit different patterns 48 (Alves et al., 2010). For instance, aboveground biomass (AGB) was reported to increase with increasing altitude 49 in tropical forests in Brazil (Alves et al., 2010) and Ethiopia (Girma et al., 2014), and in the moist temperate 50 forest in Western Himalaya (Gairola et al., 2011). For the C storage of soil in the forest ecosystems, there has 51 been no consistent altitudinal patterns (Vitousek et al., 1992; Garten and Hanson, 2006; Zhu et al., 2010; Girma 52 et al., 2014). Despite the abovementioned studies on altitudinal patterns of the C storage in forest ecosystems, 53 most of them just explored parts of the ecosystem C pools (e.g., Kakubari, 1991; Alves et al., 2010), or ignored 54 the differences in stand age and forest types (e.g., Zhu et al., 2010), which might vary substantially with altitude 55 (Zhang et al., 2009; Alves et al., 2010). In some regions, forests are more likely to be disturbed by human 56 activities at lower altitudes, usually resulting in younger forests, and thus less C accumulation (Zhang et al., 57 2009; Alves et al., 2010). Generally speaking, in aging forest stands, the C storage of both the ecosystem and 58 vegetation tended to increase (Gower et al., 1997; Pregitzer and Euskirchen, 2004; Zhang et al., 2009; Zhu et 59 al., 2017b), while that of the soil, woody debris, and litter varied as well (Gower et al., 1997; Pregitzer and 60 Euskirchen, 2004; Peichl & Arain, 2006; Nave et al., 2010; Li et al., 2011). To date, very few studies have 61 focused on the integrative effects of altitude and stand age (or disturbance) on C storage and its distribution in 62 forest ecosystems (Zhang et al., 2009; Alves et al., 2010).

Besides, previous studies have focused on either tropical or temperate forest ecosystems, with much less attention given to subtropical forests, which are unneglectable in global C cycling. The net ecosystem productivity (NEP) of East Asian monsoon subtropical forests was estimated to account for 30% of the total NEP of Asian forests (Yu et al., 2014). The beech (*Fagus* L., Fagaceae) forests are widespread in temperate and subtropical mountain regions of China (Cao, 1995). As representative trees in temperate broadleaved forests in the Northern Hemisphere (Fang and Lechowicz, 2006), the biomass and C storage of beech forests have been

Page 4

extensively explored in Europe (F. sylvatica), Japan (F. crenata) and the USA (F. grandifolia) (e.g., Mund, 70 2004; Poivesan et al., 2005; Takadi, 1969; Martin and Bailey, 1999; Jenkins et al., 2001), both on local 71 (Kakubari, 1991; Mund, 2004) and regional scales (Poivesan et al., 2010). However, the C pools of beech forests 72 in China have seldom been studied (Zhou et al., 2018). On the southeast slope of Mt. Fanjingshan, there is a 73 concentrated and consecutive distribution of beech forests over a wide range of altitude (1000-2020 m), which 74 is quite unique for beech forests in China (Fei et al., 1999). To our best knowledge, there is no such a wide 75 range of altitudinal gradient on local scales in any other regions for Chinese beeches. Thus, this area is ideal for 76 exploring the C storage and distribution in Chinese beech forest ecosystems and their response to varying 77 environmental conditions on a local scale.

In the present study, nine beech forest stands were investigated along an altitudinal gradient from 1095 to 1930 m on Mt. Fanjingshan. The four components of C pools (i.e. vegetation, soil, litter and woody debris) were estimated. Here, we aimed to 1) quantify the C storages and distributions of the beech forest ecosystems along the altitudinal gradient, 2) evaluate the key driving factors of altitudinal patterns of C storage on an ecosystem level, and 3) compare the C storage and distribution patterns with beech forests worldwide.

#### 83 2 Materials and Methods

#### 84 2.1 Study sites

The present study was conducted on Mt. Fanjingshan, which is located in the northeast part of Guizhou Province, 86 Southwest China (27.78-28.02 N, 108.60-108.81 E). This region features a humid subtropical monsoon 87 climate with a mean annual temperature (MAT) of 5.0-17.0 °C, where the mean annual precipitation (MAP) is 88 approximately 1100-2600 mm. The altitudinal range of Mt. Fanjingshan is > 2000 m, and thus it has a relatively 89 complete vertical vegetation gradient. On Mt. Fanjingshan, there are representative beech forests in subtropical 90 China, which are consecutive and cover a relatively large area (Editorial Board of the Scientific Survey of the 91 Fanjingshan Mountain Preserve Guizhou Province, China, 1986). Here, evergreen and deciduous broad-leaved 92 forests dominated by the genus Fagus are one of the main forest types, appearing at 1000–2020 m (Fei et al., 93 1999). While F. longipetiolata grows at lower altitudes in comparison to F. lucida, the latter species mainly 94 appears at 1400-1900 m. Soils in the beech forests are mountainous yellow soil or yellow-brown soil.

The study sites were located on the southeast slope of Mt. Fanjingshan. In May of 2017, nine beech forest 96 plots (600 m<sup>2</sup>) were set up at 50-150 m intervals along an altitudinal gradient from 1095 to 1930 m (27.90-97 27.91 N, 108.70-108.72 E). In the lower three plots (1095-1221 m), the dominant species were F. 98 longipetiolata, while the six plots above 1400 m were dominated by F. lucida. In addition to the genus Fagus, 99 other dominant tree species were within the genera Cyclobalanopsis, Castanopsis, and Lithocarpus of Fagaceae; 100 as well as species from the family Lauraceae (mostly evergreen species). Common species in the understory 101 shrub layer are Yushania brevipaniculata and species from Theaceae, Lauraceae, Symplocaceae, Ericaceae, and 102 Aquifoliaceae, while the herb layer is dominated by Pteridium spp., Carex spp., and Viola spp.

Of the nine beech plots, the lower two plots were secondary forests (disturbed by fires in the 1970's), and104 the other seven plots were primary forests. Detailed information of the plots is listed in Table 1.

In the present study, plots were not replicated as it is difficult to find stands with similar communitystructure and environmental conditions at each altitude (Li et al., 2011).

#### 107 2.2 Field investigations

CC D

Page 5

- Basic site information was recorded at each plot (i.e. latitude, longitude, altitude, slope, aspect, forest origin,
- and disturbance type). The C densities (storage per unit area, Mg C ha<sup>-1</sup>) of vegetation (trees, shrubs, and herbs),
- woody debris, litter, and soil were investigated and sampled respectively.

#### 111 2.2.1 Vegetation carbon storage

The DBH (cm) and height (m) were measured for all living trees with a diameter at breast height (DBH)  $\geq 3$ 113 cm within each plot. Each plot (600 m<sup>2</sup>) was divided into six subplots (10  $\times$  10 m in size) and shrubs (including 114 young trees [DBH < 3 cm]) from two subplots were investigated in detail to obtain the number, base diameter 115 of the stem, height, and coverage of each species. Allometric equations for trees and shrubs in near regions were 116 used to calculate the biomass of each species in each plot (Tables S1, 2), and the biomass of shrubs was 117 calculated using the mean values of the two subplots. Herbs from  $1 \times 1$  m subplots were harvested and weighed 118 after oven drying at 65 °C to a constant weight. Subsequently, with the above- and belowground parts treated 119 separately, the oven-dried herb samples were ground for C analysis using an Elemental Analyser (2400 II CHN 120 Elemental Analyser; Perkin-Elmer, Boston, MA, USA) (Zhu et al., 2015). For trees and shrubs, an assumed 121 factor of 0.5 was adopted to convert live biomass to C content (Myneni et al., 2001).

### 122 2.2.2 Woody debris and litter carbon storage

In this study, woody debris was divided into coarse woody debris (CWD) and fine woody debris (FWD). CWD 124 was defined as dead wood with a diameter  $\geq 10$  cm at the larger end, including standing snags and fallen logs. 125 FWD was regarded as dead wood with a diameter of 2 to 10 cm (Zhu et al., 2017a). In each plot (600 m<sup>2</sup>), DBH 126 (cm) and height (m) were measured for all standing snags, while the lengths, diameters of middle and both ends, 127 and decay degrees (1-4) of fallen logs were also measured (Zhu et al., 2017a). CWD samples with different 128 decay degrees were selected, and for each log, three sections (10-20 cm length discs) were weighed after oven 129 drying at 85 °C to a constant weight and the volumes were also measured. The ratio of oven-dried weight to the 130 volumes of the CWD discs was extrapolated to estimate the biomass of CWD with different decay degrees in 131 the plot. The volumes of standing snags and fallen logs were calculated using Eq. (1):

$$Volume = \frac{\pi d^2 L}{4}$$
(1)

Where d (cm) is the average diameter of the fallen logs or the DBH of standing snags, and L (m) is the length
or height of the logs. In addition, the C content of the CWD samples was also determined using an elemental
analyser.

Within each plot (600 m<sup>2</sup>), FWD from the forest floor was collected and weighed. Three 1 × 1 m subplots were randomly set up for litter investigation and all litter (including fallen leaves, small dead wood with a diameter of < 2 cm, and other plant debris) within the subplots was collected and weighed. To determine the C content of FWD and litter, three samples of each were obtained and weighed after oven drying at 65 °C for 48 hours, and subsequently ground and sieved (1 mm).</p>

#### 140 2.2.3 Soil carbon storage

141Three replicated soil profiles were randomly sampled within each plot (600 m²). And in the nine plots, the142deepest soil depth was 50 cm. Thus, soils were sampled separately at four depths (0–10, 10–20, 20–30, and 30–

143 50 cm). The soil bulk density was estimated using a 100 cm<sup>3</sup> standard container (50.5 mm in diameter and 50

144 mm in height), and soil samples were oven-dried at 105 °C to a constant weight to measure soil gravimetric

 $\odot$ 

Page 6

- moisture. At each depth, a second soil sample (approx. 300 g) was taken for C content analysis. The samples
- were air dried at room temperature (approx. 25 °C), rocks and plant debris removed, and subsequently ground 147
- and sieved (0.15 mm) for the determination of C content.
- In each plot, the ages of ten beech trees with the relatively largest DBH values were determined by tree
- ring analysis, and the oldest age of the ten trees was used to represent the stand age (Worbes et al., 2003).

#### 150 2.3 Statistical Analyses

The relationships between different components of the ecosystem C pools and altitude, as well as the stand age,

- were plotted using linear regression analyses. A one-way analysis of variance (ANOVA) and the least
- significant difference post hoc test (LSD) were conducted to compare the differences in soil C storage among
- the nine plots and different soil depth, and the C pools of beech forests on Mt. Fanjingshan and other regions.
- 3 Results

#### 156 3.1 Vegetation carbon storage

The C storage of vegetation of the beech forests ranged from 64.4 to 364.3 Mg C ha<sup>-1</sup>. Across the nine forests, 158 the tree layer accounted for the dominant proportion of the vegetation C (63.5–360.7 Mg C ha<sup>-1</sup>, accounting for 159 98.3–99.2%), which was substantially larger than the shrub (0.6–4.6 Mg C ha<sup>-1</sup>, 0.5–1.7%) and herb layers 160 (0.03–0.35 Mg C ha<sup>-1</sup>, 0.01–0.3%) (Figure 1a; Table 2). Vegetation C storage of the beech forests increased 161 significantly as altitude increased from 1095 m to 1930 m (F = 31.9, P < 0.001). The C storage of both the tree 162 and shrub layers tended to increase significantly with increased altitude ( $R^2 = 0.82$ , P < 0.001;  $R^2 = 0.64$ , P < 0.001;  $R^2 = 0.001$ ;  $R^2 = 0$ 163 0.01, respectively), while the herb layer exhibited no significant variation (Figure 1a). Furthermore, the 164 aboveground biomass C storage accounted for the majority of vegetation C storage (54.3-270.2 Mg C ha<sup>-1</sup>, 165 accounting for 73.7-84.4%), and both the above- and belowground biomass C storage increased significantly 166 as altitude increased ( $R^2 = 0.83$ , P < 0.001;  $R^2 = 0.79$ , P = 0.001, respectively; Figure 1b).

#### 167 3.2 Carbon storage in litter and woody debris

Litter C storage varied between 0.7 and 1.5 Mg C ha-1 across the nine forests, while the C storage of woody 169 debris was 0.2–14.6 Mg C ha<sup>-1</sup>. The C storage of woody debris was mainly attributed to CWD, which composed 170 over 90% of the total woody debris C storage (1.0-14.6 Mg C ha<sup>-1</sup>, 92.5-96%) in most plots, with the exception 171 of three plots that varied between 41.8-62.5% (0.2-0.7 Mg C ha-1). As altitude increased, the C storage of 172 woody debris increased significantly (F = 18.9, P = 0.003), mainly owing to the increase in CWD ( $R^2 = 0.73$ , 173 P = 0.003; Figure 2a). The C storage of FWD exhibited an increasing trend despite statistically insignificant ( $R^2$ 174 = 0.43, P = 0.056; Figure 2a), while litter C storage exhibited no significant altitudinal patterns (Figure 2a). 175 The relationship between the C storage of plant debris and vegetation was further explored. Vegetation C

- storage appeared to exert no significant effect on litter C storage (Figure 2b). For woody debris, there was a
- positive but insignificant response to the increase in vegetation C storage ( $R^2 = 0.40$ , P = 0.067). CWD also 178 showed a slight but insignificant increasing trend ( $R^2 = 0.41$ , P = 0.06), while FWD exhibited no significant 179 variation as vegetation C storage increased (Figure 2b).

#### 180 3.3 Carbon storage in soils

- Soil C storage in the nine forests ranged from 88.3 ±2.0 to 229.7 ±81.3 Mg C ha<sup>-1</sup>, and with the exception of the
- relatively higher value of 229.7  $\pm$  81.3 Mg C ha<sup>-1</sup> from the plot at 1136 m, soil C storage in the other plots

Page 7

183varied between  $88.3 \pm 2.0$  and  $124.8 \pm 19.5$  Mg C ha<sup>-1</sup> (Table 2). Across the nine plots, the contribution of the184upper three soil layers (0–10, 10–20, 20–30 cm) were similar and significantly larger than that in the deepest185layer (30–50 cm) (P < 0.001), but the distribution patterns of C storage along the soil depth varied at different186altitude (Table 2). As altitude increased, C storage in the second layer (10–20 cm) decreased slightly (F = 5.8,187P = 0.02), while the total soil C storage and C storages of other vertical layers (0–10, 20–30, 30–50 cm) all188exhibited no significant trends (Table 2). The effects of the other C components (vegetation, woody debris and189litter) on soil C storage were also explored, but no significant effects were observed.

#### 190 3.4 Distribution of ecosystem carbon storage in the four carbon components

The ecosystem C storage of the nine forests varied between 190.7 and 503.9 Mg C ha<sup>-1</sup>, where the dominant 192 contributors were vegetation (accounting for 33.7-73.9%) and soil (24.6-65.4%). The contribution of woody 193 debris and litter C storage was relatively lower (0.25-3.4%) in comparison to vegetation and soil, where only 194 0.05-3.1% could be attributed to woody debris and 0.2-0.7% to litter (Table 2). The ecosystem C storage 195 increased significantly with increased altitude (F = 9.7, P = 0.02; Table 2). The patterns of contribution to 196 ecosystem C storage also changed with increasing altitude, where the contribution of vegetation C storage was 197 33.7% at 1095 m, subsequently increased to 73.9% at 1580 m, and remained stable thereafter (71.5-72.5%) 198 (Table 2). In contrast, the contribution of soil to ecosystem C storage decreased with increasing altitude, which 199 was around two-thirds (65.4%) in the lowest forest, and declined to one quarter (24.6%) of the total C storage 200 in the highest plot (Table 2). The contribution of litter C storage exhibited no significant altitudinal trends, while 201 that of woody debris C storage increased in fluctuation from 0.4% to 2.9% with increasing altitude (Table 2).

#### 202 4 Discussion

# 4.1 Contribution patterns of carbon components of the ecosystems and comparisons with beech forests worldwide

205 Mt. Fanjingshan is quite unique and ideal for studies of Chinese beech forests as it has the widest altitudinal 206 range of Chinese beech forests at a local scale of any region. The C storages of different C pools in the beech 207 forest ecosystems on Mt. Fanjingshan were comparable to those with similar range of stand age in other 208 countries and regions worldwide (Table 3; Figure 3). The total ecosystem C storage averaged 335 Mg C ha<sup>-1</sup> 209 (ranging from 190.7 to 503.9 Mg C ha<sup>-1</sup>) on Mt. Fanjingshan, exhibiting no significant difference with that of 210 beech forests in Europe (averaging 308 Mg C ha<sup>-1</sup>) and Mt. Yueliangshan in southeast China (averaging 319 211 Mg C ha<sup>-1</sup>) (P > 0.1; Table 3). And the C storages of vegetation, soil and plant debris were also similar to that 212 of Europe, Mt. Yueliangshan, or Japan (P > 0.05; Table 3). According to Pregitzer and Euskirchen (2004), the 213 average ecosystem C storage of temperate forests worldwide with different stand ages ( $\leq 200$  years old) ranged 214 from 121 to 537 Mg C ha<sup>-1</sup>, also close to the values observed in this study.

Herein, the distribution patterns of C storage of beech forests were also comparable to those of beech forests in Europe and Mt. Yueliangshan (Table 3), as well as other types of temperate and subtropical forests in China (e.g., Niu et al., 2009; Zhang and Wang, 2010; Zhu et al., 2017b). In these forests, the accumulation of ecosystem C storage was mainly attributed to vegetation and soil C, and the contribution of plant debris was relatively minor (

Page 8

2011), possibly because of the differences in stand history, disturbance regime and the standards of standselection (Zhu et al. 2017b).

#### 225 4.2 Altitudinal patterns of carbon storage in the beech forest ecosystems

In the present study, the C storage of vegetation was observed to increase with increasing altitude (1095– 227 1930 m) on Mt. Fanjingshan, differing from the general decreasing tendency (e.g., Vitousek et al., 1992; 228 Leuschner et al., 2007; Zhu et al., 2010) or the hump-shaped variation pattern as observed in the F. crenata 229 forests in the Naeba Mountains (550-1500 m) in central Japan (Satoo, 1973; Kakubari, 1991). Nevertheless, 230 similar increasing trends have also been reported in some tropical and temperate mountain forests worldwide 231 (e.g., Zhang et al., 2009; Alves et al., 2010; Gairola et al., 2011; Girma et al., 2014), and the stand age or 232 disturbance regime was found to be one of the key drivers (Zhang et al., 2009; Alves et al., 2010). Thus, we 233 further investigated the stand age of beech forests on Mt. Fanjingshan and explored its relationships with the C 234 storage of different C pools. Here, the stand age tended to increase with increasing altitude ( $R^2 = 0.56$ , P = 0.02; 235 Figure 4a), as plots at lower altitudes have suffered from more human disturbance. And vegetation C storage 236 was found to concomitantly increase with stand age ( $R^2 = 0.72$ , P = 0.004; Figure 4b). The positive effects of 237 stand age on vegetation C storage have also been reported in many studies on boreal, temperate, and subtropical 238 natural forests and plantations (Pregitzer and Euskirchen, 2004; Peichl and Arain, 2006; Bradford and 239 Kastendick, 2010; Zhu et al., 2017b). And for beech forests worldwide, vegetation C storage also tended to 240 increase with increasing stand age (Figure 5b). Thus, stand age was possibly one of the key driving factors of 241 the altitudinal changes in vegetation C storage in the beech forests on Mt. Fanjingshan.

The C storage of woody debris exhibited positive altitudinal tendency, while that of litter showed no patterns, 243 and both showed no significant relationships with stand age (Figure 4b). Different stand age patterns have been 244 reported in some previous studies (Pregitzer and Euskirchen, 2004; Jandl et al., 2007; Bradford and Kastendick, 245 2010) and in beech forests worldwide (Figure 5b, c), while there were similar results as well (Zhu et al., 2010; 246 Zhu et al., 2017a). The C storage of plant debris was mainly controlled by aboveground biomass (input) and 247 the rate of decomposition (output) (Zhu et al., 2017a). Herein, vegetation C storage tended to increase with 248 increasing altitude, resulting in the increased input of plant debris. At the same time, decreased air temperature 249 led to the slower decomposition rate, and thus the lower output. Therefore, the C storage of woody debris 250 exhibited an increasing altitudinal pattern. However, the lack of a significant relationship between the C storage 251 of litter and altitude or stand age is likely a result of the relatively faster decomposition rate of litter, thereby 252 facilitating a balance between the input and output (Zhu et al., 2017a).

In this study, the C storage of soil also exhibited no significant patterns in relation to altitude (Figure 2) or 254 stand age (Figure 4b). Previous studies have demonstrated that there was no consistent response of soil C storage 255 to changes in altitude, which has been shown to increase (Garten and Hanson, 2006; Zhu et al., 2010), decrease 256 (Vitousek et al., 1992), or remain relatively stable (Vitousek et al., 1992). Soil C storage usually tended to 257 accumulate in aging forests (Hooker & Compton, 2003; Pregitzer and Euskirchen, 2004), but it will be affected 258 by numerous factors such as previous land use, climate and vegetation types (Paul et al. 2002; Peichl and Arain, 259 2006). Therefore, with increasing stand age, the U-shaped variation or no significant trends of soil C storage 260 have also been observed (Paul et al. 2002; Peltoniemi et al., 2004). In beech forests worldwide, there was no 261 significant relationships between stand age and soil C storage as well (Figure 5e). The insignificant variation of 262 soil C storage to altitude or stand age on Mt. Fanjingshan was probably related to the previous land use and

Page 9

disturbance of the beech forests at lower altitudes. As abovementioned, the lower two beech forests were postfire secondary forests, which might have accumulated large quantities of C in soil before the dramatic disturbance (Paul et al., 2002; Nave et al., 2010). In addition, the input and output of soil organic C are affected by numerous factors, and plots at lower altitudes generally suffered more from human disturbance (Pregitzer and Euskirchen, 2004; Zhang et al., 2009). Therefore, soil C storages in younger beech forests at lower altitudes showed no significant difference to those in older forests at higher altitudes.

The ecosystem C storage of the beech forests also increased with increasing altitude, and was mainly 270 attributed to increases in vegetation C, while the contribution of soil C storage declined concomitantly. As 271 abovementioned, the stand age tended to increase as altitude increased, and increasing stand age led to a slight 272 despite insignificant increase in ecosystem C storage (Figure 4). Similar age patterns have also been observed 273 in beech forests worldwide (Figure 5e, f). It has been widely reported that the C storages of both the ecosystem 274 and vegetation tended to increase in aging forest stands (Gower et al., 1997; Pregitzer and Euskirchen, 2004; 275 Zhu et al., 2017a, b), and in older forests, the majority of ecosystem C storage was attributed to vegetation, 276 while soil was the dominant contributor of C storage in younger forests (Pregitzer and Euskirchen, 2004; Zhang 277 and Wang, 2010).

#### 278 4.3 Potential limitations

The C storage and contributions of different ecosystem components in nine beech forests were fully investigated and estimated along an altitudinal gradient on Mt. Fanjingshan. However, some limitations of the sampling and analyses may influence the accuracy of the results. For example, it was difficult to replicate the plots owing to the conditions of the study site (Li et al., 2011). Therefore, the limited quantities of samples could lead to bias in model-based analyses. Furthermore, the estimation of the C storage of trees and shrubs using species-specific allocation equations and the broadly-applied conversion factor of 0.5 used to estimate the C concentration could have resulted in estimation errors (Li et al., 2011; Ma et al., 2018).

#### 286 5 Conclusions

The present study investigated the C storage and contributions of different ecosystem components 288 (vegetation, woody debris, litter, and soil) in nine beech forests along an altitudinal gradient on Mt. Fanjingshan. 289 The results show that the ecosystem C storage of the beech forests ranged from 190.7 to 503.9 Mg C ha<sup>-1</sup>, and 290 was mainly attributed to vegetation C (from 64.4 to 364.3 Mg C ha<sup>-1</sup>, accounting for 33.7–73.9%) and soil C 291 (from 88.3  $\pm 2.0$  to 229.7  $\pm 81.3$  Mg C ha<sup>-1</sup>, 24.6–65.4%). No more than 4% of the ecosystem C storages were 292 stored in woody debris (0.25–3.4%) and litter (0.2–0.7%). The values of C storage and the distribution patterns 293 of beech forests on Mt. Fanjingshan are comparable to that in other regions worldwide. The ecosystem C storage 294 increased significantly with increasing altitude. In regards to components of the forest ecosystems, the C storage 295 in vegetation and woody debris increased concomitantly with altitude, while that of litter and soil showed no 296 significant variations. The stand age was found to be one of the key drivers of such altitudinal patterns, 297 especially for vegetation C storage. Not only does the present study provide reliable data for understanding the 298 structure and function of beech forests in China, it also suggests that the effects of stand age and previous land 299 use or disturbance should be given more weight in studies regarding forest ecosystem C storage. In the future, 300 more detailed and comprehensive surveys will be indispensable in enhancing the accuracy of estimations, where 301 other factors except for climatic factors and stand age also deserve more consideration.

Biogeo

Page 10

- Data availability. Data used in this study can be found in the Supplement.
- **Competing interests.** The authors declare that they have no conflict of interest.
- Acknowledgments. This work was supported by National Science and Technology Basic Project of China
- (2015FY210200, 2017YFA0605101) and National Natural Science Foundation of China (31700374,
- 31621091). And we thank Huifeng Hu and Suhui Ma for helpful discussions.

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

Biogeosciences Discussions

| 347<br>348<br>349<br>350 | <ul> <li>Harmon, M. E., Krankina, O. N., Yatskov, M., and Matthews, E.: Predicting broad-scale carbon stores of woody detritus from plot-level data, in: Assessment Methods for Soil Carbon, Lai, R., Kimble, J., Stewart, B. A., CRC Press, New York, NY, USA, 533–552, 2001.</li> <li>He, Y. J., Qin, L., Li, Z. Y., Liang, X. Y., Shao, M. X., and Tan, L.: Carbon storage capacity of monoculture</li> </ul> |
|--------------------------|------------------------------------------------------------------------------------------------------------------------------------------------------------------------------------------------------------------------------------------------------------------------------------------------------------------------------------------------------------------------------------------------------------------|
| 351                      | and mixed-species plantations in subtropical China, For. Ecol. Manag., 295, 193-198, 2013.                                                                                                                                                                                                                                                                                                                       |
| 352                      | Hu, X., Zhu, J., Wang, C., Zheng, T., Wu, Q., Yao, H., and Fang, J.: Impacts of fire severity and post-fire                                                                                                                                                                                                                                                                                                      |
| 353                      | reforestation on carbon pools in boreal larch forests in Northeast China, J. Plant. Ecol., 9, 1–9, 2015.                                                                                                                                                                                                                                                                                                         |
| 354                      | IPCC: Summary for Policymakers, in: Climate Change 2013: The Physical Science Basis. Contribution of                                                                                                                                                                                                                                                                                                             |
| 355                      | Working Group I to the Fifth Assessment Report of the Intergovernmental Panel on Climate Change, edited                                                                                                                                                                                                                                                                                                          |
| 356                      | by: Stocker, T. F., Qin, D., Plattner, GK., Tignor M., Allen, S. K., Boschung, J., Nauels, A., Xia, Y., Bex,                                                                                                                                                                                                                                                                                                     |
| 357                      | V., and Midgley, P. M., Cambridge University Press, Cambridge, United Kingdom and New York, NY,                                                                                                                                                                                                                                                                                                                  |
| 358                      | USA, 2013.                                                                                                                                                                                                                                                                                                                                                                                                       |
| 359                      | Hooker, T. D. and Compton, J. E.: Forest ecosystem carbon and nitrogen accumulation during the first century                                                                                                                                                                                                                                                                                                     |
| 360                      | after agricultural abandonment, Ecol. Appl., 13, 299-313, 2003.                                                                                                                                                                                                                                                                                                                                                  |
| 361                      | Jandl, R., Lindner, M., Vesterdal, L., Bauwens, B., Baritz, R., Hagedorn, F., Johnson, D. W., Minkkinen, K.,                                                                                                                                                                                                                                                                                                     |
| 362                      | and Byrne, K. A.: How strongly can forest management influence soil carbon sequestration?. Geoderma,                                                                                                                                                                                                                                                                                                             |
| 363                      | 137, 253–268, 2007.                                                                                                                                                                                                                                                                                                                                                                                              |
| 364                      | Jenkins, J. C., Birdsey, R. A., and Pan, Y.: Biomass and NPP estimation for the Mid-Atlantic region (USA)                                                                                                                                                                                                                                                                                                        |
| 365                      | using plot-level forest inventory data, Ecol. Appl., 11, 1174-1193, 2001.                                                                                                                                                                                                                                                                                                                                        |
| 366                      | Kakubari, Y.: Primary productivity changes for a fifteen-year period in a natural beech (Fagus crenata) forest                                                                                                                                                                                                                                                                                                   |
| 367                      | in the Naeba mountains, J. Jpn. For. Soc., 73, 370-374, 1991.                                                                                                                                                                                                                                                                                                                                                    |
| 368                      | Körner, C.: The use of 'altitude' in ecological research, Trends Ecol. Evol., 22, 569-74, 2007.                                                                                                                                                                                                                                                                                                                  |
| 369                      | Leuschner, C., Moser, G., Bertsch, C., Röderstein, M., and Hertel, D.: Large altitudinal increase in tree                                                                                                                                                                                                                                                                                                        |
| 370                      | root/shoot ratio in tropical mountain forests of Ecuador, Basic Appl. Ecol., 8, 219-230, 2007.                                                                                                                                                                                                                                                                                                                   |
| 371                      | Li, X., Yi, M. J., Son, Y., Park, P. S., Lee, K. H., Son, Y. M., Jeong, M. J.: Biomass and carbon storage in an                                                                                                                                                                                                                                                                                                  |
| 372                      | age-sequence of Korean pine (Pinus koraiensis) plantation forests in central Korea, J. Plant Biol., 54, 33-                                                                                                                                                                                                                                                                                                      |
| 373<br>374<br>375<br>376 | <ul><li>42, 2011.</li><li>Ma, S. H., He, F., Tian, D., Zou, D. T., Yan, Z. B., Yang, Y. L., Zhou, T. C., Huang K. Y., Shen, H. H., and Fang, J. Y.: Variations and determinants of carbon content in plants: a global synthesis, Biogeosciences, 15, 693, 2018.</li></ul>                                                                                                                                        |
| 377                      | Martin, C. W. and Bailey, A. S.: Twenty years of change in a northern hardwood forest, Forest Ecol.                                                                                                                                                                                                                                                                                                              |
| 378<br>379<br>380        | <ul><li>Manag., 123, 253–260, 1999.</li><li>Martin, J. L., Gower, S. T., Plaut, J., and Holmes, B.: Carbon pools in a boreal mixedwood logging chronosequence, Glob. Chang. Biol., 11, 1883–1894, 2005.</li></ul>                                                                                                                                                                                                |
| 381                      | Mund, M.: Carbon pools of European beech forests (Fagus sylvatica) under different silvicultural management,                                                                                                                                                                                                                                                                                                     |
| 382<br>383<br>384<br>385 | Ph.D. thesis, The University of G \u00e4tingen, Germany, 256 pp., 2004. Myneni, R. B., Dong, J., Tucker, C. J., Kaufmann, R. K., Kauppi, P. E., Liski, J., Zhou, L., Alexeyev, V., and Hughes, M. K.: A large carbon sink in the woody biomass of northern forests, P. Natl. Acad. Sci. USA, 98, 14784–14789, 2001.                                                                                              |
| 386                      | Nave, L. E., Vance, E. D., Swanston, C. W., and Curtis, P. S.: Harvest impacts on soil carbon storage in                                                                                                                                                                                                                                                                                                         |
| 387<br>388<br>389<br>390 | <ul> <li>temperate forests, Forest Ecol. Manag., 259, 857–866, 2010.</li> <li>Niu, D., Wang, S. L., and OuYang, Z. Y.: Comparisons of carbon storages in <i>Cunninghamia lanceolata</i> and <i>Michelia macclurei</i> plantations during a 22-year period in southern China, J. Environ. Sci-China, 21, 801–805, 2009.</li> </ul>                                                                                |

| 391                                           | Pan, Y. D., Birdsey, R. A., Fang, J. Y., Houghton, R., Kauppi, P. E., Kurz, W. A., Phillips, O. L., Shvidenko,                                                                                                                                                                                                                                                                                                                                                                                                                                                                                                                                                                                                                                |
|-----------------------------------------------|-----------------------------------------------------------------------------------------------------------------------------------------------------------------------------------------------------------------------------------------------------------------------------------------------------------------------------------------------------------------------------------------------------------------------------------------------------------------------------------------------------------------------------------------------------------------------------------------------------------------------------------------------------------------------------------------------------------------------------------------------|
| 392                                           | A., Lewis, S. L., Canadell, J. G., Ciais, P., Jackson, R. B., Pacala, S. W., Piao, S., Rautiainen, A., Sitch, S.,                                                                                                                                                                                                                                                                                                                                                                                                                                                                                                                                                                                                                             |
| 393                                           | and Hayes, D.: A large and persistent carbon sink in the world's forests, Science, 333, 988-993, 2011.                                                                                                                                                                                                                                                                                                                                                                                                                                                                                                                                                                                                                                        |
| 394                                           | Paul, K. I., Polglase, P. J., Nyakuengama, J. G., Khanna, P. K.: Change in soil carbon following afforestation,                                                                                                                                                                                                                                                                                                                                                                                                                                                                                                                                                                                                                               |
| 395<br>396<br>397<br>398<br>399               | <ul> <li>Forest Ecol. Manag., 168, 241–257, 2002.</li> <li>Peichl, M. and Arain, M. A.: Above-and belowground ecosystem biomass and carbon pools in an age-sequence of temperate pine plantation forests, Agr. Forest Meteorol., 140, 51–63, 2006.</li> <li>Peltoniemi, M., M äkip ää, R., Liski, J., and Tamminen, P.: Changes in soil carbon with stand age-an evaluation of a modelling method with empirical data, Glob. Chang. Biol., 10, 2078–2091, 2004.</li> </ul>                                                                                                                                                                                                                                                                    |
| 400                                           | Piovesan, G., Alessandrini, A., Baliva, M., Chiti, T., D'Andrea, E., De Cinti, B., Di Filippo, A., Hermanin, L.,                                                                                                                                                                                                                                                                                                                                                                                                                                                                                                                                                                                                                              |
| 401                                           | Lauteri, M., Scarascia-Mugnozza, G., Schirone, B., Ziaco, E., Matteucci, G.: Structural patterns, growth                                                                                                                                                                                                                                                                                                                                                                                                                                                                                                                                                                                                                                      |
| 402                                           | processes, carbon stocks in an Italian network of old-growth beech forests, Ital. J. Forest Mt. Environ., 65,                                                                                                                                                                                                                                                                                                                                                                                                                                                                                                                                                                                                                                 |
| 403                                           | 557–590, 2010.                                                                                                                                                                                                                                                                                                                                                                                                                                                                                                                                                                                                                                                                                                                                |
| 404                                           | Piovesan, G., Di Filippo, A., Alessandrini, A. E. A., Biondi, F., and Schirone, B.: Structure, dynamics and                                                                                                                                                                                                                                                                                                                                                                                                                                                                                                                                                                                                                                   |
| 405<br>406<br>407                             | <ul> <li>dendroecology of an old-growth <i>Fagus</i> forest in the Apennines, J. Veg. Sci., 16, 13–28, 2005.</li> <li>Pregitzer, K. S. and Euskirchen, E. S.: Carbon cycling and storage in world forests: Biome patterns related to forest age, Glob. Chang. Biol., 10, 2052–2077, 2004.</li> </ul>                                                                                                                                                                                                                                                                                                                                                                                                                                          |
| 408                                           | Satoo, T.: A synthesis of studies by the harvest method: primary production relations in the temperate deciduous                                                                                                                                                                                                                                                                                                                                                                                                                                                                                                                                                                                                                              |
| 409                                           | forests of Japan, in: Analysis of Temperate Forest ecosystems, Reichle, D. E., Springer, Berlin, Heidelberg,                                                                                                                                                                                                                                                                                                                                                                                                                                                                                                                                                                                                                                  |
| 410                                           | 55–72, 1973.                                                                                                                                                                                                                                                                                                                                                                                                                                                                                                                                                                                                                                                                                                                                  |
| 411                                           | Tadaki, Y., Hatiya, K., and Tochiaki, K.: Studies on the Production Structure of Forest (XV), J. Jpn. For. Soc.,                                                                                                                                                                                                                                                                                                                                                                                                                                                                                                                                                                                                                              |
| 412<br>413<br>414                             | <ul> <li>51, 331–339, 1969[In Japanese with English abstract].</li> <li>Vitousek, P. M., Aplet, G., Turner, D., and Lockwood, J. J.: The Mauna Loa environmental matrix: foliar and soil nutrients, Oecologia, 89, 372–382, 1992.</li> </ul>                                                                                                                                                                                                                                                                                                                                                                                                                                                                                                  |
| 415                                           | Woodall, C. W. and Liknes, G. C.: Relationships between forest fine and coarse woody debris carbon stocks                                                                                                                                                                                                                                                                                                                                                                                                                                                                                                                                                                                                                                     |
| 416                                           | across latitudinal gradients in the United States as an indicator of climate change effects, Ecol. Indic., 8,                                                                                                                                                                                                                                                                                                                                                                                                                                                                                                                                                                                                                                 |
| 417                                           | 686–690, 2008.                                                                                                                                                                                                                                                                                                                                                                                                                                                                                                                                                                                                                                                                                                                                |
| 418<br>419<br>420<br>421<br>422<br>423<br>424 | <ul> <li>Worbes, M., Staschel, R., Roloff, A., and Junk, W. J.: Tree ring analysis reveals age structure, dynamics and wood production of a natural forest stand in Cameroon, For. Ecol. Manag., 173, 105–123, 2003.</li> <li>Yang, Y. H., Fang, J. Y., Datta, A., Li, P., Ma, W. H., Mohammat A., Shen H. H., Hu H. F., Knapp, B. O., and Smith, P.: Stoichiometric shifts in surface soils over broad geographical scales: evidence from China's grasslands, Glob. Ecol. Biogeogr., 23, 947–955, 2014.</li> <li>Yang, Y. H., Fang, J. Y., Tang, Y. H., Ji, C. J., Zheng, C. Y., He, J. S., and Zhu, B.: Storage, patterns and controls of soil organic carbon in the Tibetan grassland, Glob. Chang. Biol., 14, 1592–1599, 2008.</li> </ul> |
| 425                                           | Yu, G., Chen, Z., Piao, S., Peng, C., Ciais, P., Wang, Q., Li, X., and Zhu, X.: High carbon dioxide uptake by                                                                                                                                                                                                                                                                                                                                                                                                                                                                                                                                                                                                                                 |
| 426                                           | subtropical forest ecosystems in the East Asian monsoon region. P. Natl. Acad. Sci. USA, 111, 4910–4915,                                                                                                                                                                                                                                                                                                                                                                                                                                                                                                                                                                                                                                      |
| 427<br>428<br>429<br>430<br>431               | <ul> <li>2014.</li> <li>Zhang, Q. Z. and Wang, C. K.: Carbon density and distribution of six Chinese temperate forests, Sci. China Life Sci., 53, 831–840, 2010.</li> <li>Zhang, X. P., Wang, M. B., and Liang, X. M.: Quantitative classification and carbon density of the forest vegetation in Lu 'liang Mountains of China, Plant Ecol., 201, 1–9, 2009.</li> </ul>                                                                                                                                                                                                                                                                                                                                                                       |
| 432                                           | Zhou, X., Cai, Q., Xiong, X., Fang, W., Zhu, J. X., Zhu, J. L., Fang, J., and Ji, C.: Ecosystem carbon storage in                                                                                                                                                                                                                                                                                                                                                                                                                                                                                                                                                                                                                             |
| 433                                           | successional Fagus lucida forests in southwestern China, Chinese J. Plant Ecol., 2018 [In press and in                                                                                                                                                                                                                                                                                                                                                                                                                                                                                                                                                                                                                                        |
| 434<br>435<br>436                             | Chinese with English abstract].<br>Zhu, B; Wang, X. P., Fang, J. Y., Piao, S. L., Shen, H. H., Zhao, S. Q., and Peng, C. H.: Altitudinal changes in<br>carbon storage of temperate forests on Mt Changbai, northeast China, J. Plant Res., 123, 439–452, 2010.                                                                                                                                                                                                                                                                                                                                                                                                                                                                                |