# Peer review of "Changes in carbon stocks of Fagus forest ecosystems along 1"

_Biogeosciences, 2018_

## Referee Comment (RC1) · Anonymous Referee #1 · 22 Jun 2018

Review MS No.: bg-2018-242 "Changes in carbon stocks of Fagus forest ecosystems along along an extended elevation transect dominated by Fagus species. C stock estimates are based on an extensive data collection allowing a complete assessment, i.e. including all relevant C pools. The aims of the study were to quantify C stocks and to identify drivers to explain the observed pattern of ecosystem C stocks along the transect. The authors alsocompared their estimated C stocks with Fagus forests in other continents based on published data.

[Figure]

General comments

The paper is overall well written with the exception of some confused references to Tables and Figures. It is within the scope of BG addressing the question of drivers of forest ecosystem C stock changes. The C stocks estimates in living biomas, dead wood, litter and soil along the transect were based on thorough and comprehensive vegetation and soil sampling and anlysis. To place the findings in the context of similar Fagus Ecosystem worldwide, a collection of published data was used. Beyond the estimation of C stocks along the transect, the paper presents no novel methods or insights in mechanisms to explain observed pattern, it can confirm existing knowledge. Given the large range of C storage in different C pools in the examined system and also in the systems used for comparison, the value of the comparison with other studies is limited. In the comparison it may have been interesting to focus and expand on differences and their causes, e.g. regarding the contribution of litter and dead wood. The roles of forest management and use intensity were only moderately addressed, particularly in the comparison with other forests. It may also have been valuable to place the results in the context of the National Forest Resource Inventory database, cf. Fang J, Chen A, Peng C, Zhao S, Ci L (2001) Changes in Forest Biomass Carbon Storage in China Between 1949 and 1998 Science 292:2320-2322 doi:10.1126/science.1058629.

Specific comments / technical corrections

Abstract, l.22: Rephrase; the wording is too general. The study presents reliable data on C storage as one among many ecosystem functions but not for understanding structure and function of Chinese beech forests.

l.29-30: The summary for policy-makers in IPCC 2013 is not an appropriate reference here. Technically, there are 5 pools since vegetation is separated into above- and below-ground parts. Consider revising the sentence and citing IPCC (2006) 2006 IPCC Guidelines for National Greenhouse Gas Inventories Volume 4 Agriculture, Forestry and Other Land Use. Available at: http://www.ipcc-

nggip.iges.or.jp/public/2006gl/vol4.html.

l.33: Delete 'even'.

l.45 and throughout the manuscript: altitudinal, altitude etc. Consider replacing with elevation, which is more appropriate term in this context; cf. McVicar T, Körner C (2013) On the use of elevation, altitude, and height in the ecological and climatological literature Oecologia 171:335-337 doi:10.1007/s00442-012-2416-7

l.50-51: grammar – 'there has been . . . pattern' or 'there have been . . .patterns'.

l.56: Consider revising: 'less C accumulation in total' since in relative terms younger stands tend to accumulate more carbon'. Also, biomass accumulation is likely to peak in a mature forest before declining again; cf. Pregitzer and Euskirchen 2004 cited in the manuscript.

l.64: Does 'unneglectable' exist, maybe revise to 'negligible' or 'insignificant'

l.90: Is 'consecutive' appropriate; consider 'continuous'.

l.149: It would have been interesting to give some indication on the variability of stand age to demonstrate how appropriate this estimate is for the primary and possibly uneven-aged forests. Or possibly a description of the presence/absence of different age cohorts.

l.160: The reference to Tab.2 is not clear in this context as it shows C cocks in the soil, which are not referred to in the preceding sentence, or in this chapter, which is about vegetation.

l.162, 166, Fig.1: R2 is presented which only tells how well the model is fitting the actual data. In addition it would be valuable to state whether the the coeffcients are different from 0. This applies also to the results presented in sections 3.2 and 3.3.

l.181: Consider inserting 'total soil C storage'.

l.194, 195, 198, 201: The information is not in Tab. 2, possibly you ar referring to Fig. 4?

l.204: Is worldwide really appropriate? The comparison of beech forests is based on European, Japanese and Chinese data. The American data are not mentioned in this section.

l. 208: Figure 3 was not previuously introduced and is probably incorrectly cited here as Fig. 5 contains the relevant information.

l.222-223: The discussion could be extended to the effect of different management practices and intensities of use. Many beech forests in Europe have been heavily used in the past and show legacies but are now often under some form of protection; cf. Mund M, Schulze E-D (2006) Impacts of forest management on the carbon budget of European beech (Fagus sylvatica) forests Allgem Forst- und Jagdzeitung 177:47-62 and also Mund 2004 cited in the manuscript.

l.234-236: Consider moving this to the results section.

l.246-249: Please clarify this sentence 'At the same time...lower output'. The study did not measure decomposition rate. What is the meaning of output, C emissions? This was not measured. Possibly rephrase to indicate that this is an hypothesis as, for example, on l.250-251.

l.247-248: Please clarify this sentence 'Herein, . . .. Input of pland debris'. It is not clear how increased C storage can result in in increasd input of plant debris. Turnover of tree or shrub was not measured, and if it was the objective to discuss this aspect, a reference to literature such as Shaozhong Wang, Zhengquan Wang, Jiacun Gu 2017. Variation patterns of fine root biomass, production and turnover in Chinese forests. Journal of Forestry Research, 28: 1185-1194 may be appropriate.

l.253: Fig. 2 does not include information on soil.

l.264 & 182: The fact that the two secondary forests were disturbed by fire may explain

the comparatively high soil C as there may be fire-derived carbon.

l.265-267: The link between the first and second subclause is not clear. The reference Pregitzer and Euskirchen is not correct here, as they do not demonstrate a relationship between disturbance intensity and elevation.

l.267-268: 'therefore' is not appropriate as this it is an hypothesis.

l.278-285: A further limitation is the unceratinty related to the application of allometric equations to estimate tree biomass. Standard deviations are presented for soil C in Tab. 2.

l.297-298: It is not clear how the study contributes to the understanding of structure and function of beech forests in China. Rephrase to something like 'C storage and distribution among pools'.

Table 1: What is the explanation of the comparatively low age of 88 years of stand FJ4 relative to the other primary forests?

Table 2: Please indicate different meanings of the letters a and b, which are used to indicate significant differences.

Tab. 3: The data for American beech forest are missing.

Figure 3: It appears that this figure is never referred to in the text; the reference to fig 3 on l.208 appears to refer to Fig. 5.

Fig.4:Consider enlarging the figure or placing legend and coefficients differently. Table S2: What are the reasons and the effect of modifying the equations? How reasonable is it to use root:shoot ratios of trees for shrubs? This may no be appropriate, cf. Mooney HA (1972) The Carbon Balance of Plants Annual Review of Ecology and Systematics 3:315-346, and should be discussed as limitation and source of error.

---

## Referee Comment (RC2) · Anonymous Referee #2 · 20 Aug 2018

The manuscript could a contribution of interest for Biogeosciences and in principle within its specific scope but it is not suitable for publication in this form. The Authors have made a great effort in collecting many data, but the experimental design is not appropriate to the proposed objectives. In addition, statistical analysis of data is poor and misused because the statistical results do not confirm what the Authors reported as main findings. The Authors have completely neglected spatial variability in both soil and forest cover. Organic carbon content in soils is strongly dependent on the soil properties and particularly, on soil texture. The Authors should provide at least the main soil properties of the nine areas to show their homogeneity. Such an implicit assumption of homogeneity cannot hold with no information on soil properties. The nine

stands are not comparable because they have different ages (ranging between 44 and 185 years), density (ranging between 1483 and 2350 stems/ha). How do the Authors think possible to evaluate the key driving factors of altitude gradient in vegetation carbon storage? To separate the elevation effect from other attributes, it is requested to have homogeneous stands in which the only variable factor is elevation.

Results are not supported by statistical analysis. The changes in vegetation carbon storage along the elevation gradient makes no sense (Fig. 1). A simple visual inspection of Fig. 1a for trees data, shows a regression line through two cluster of points and reporting significant coefficient of determination has no statistical meaning. The same occurred for the aboveground vegetation (Fig. 1b). Shrub and herb have no gradient (Fig, 1a): the regression line is almost horizontal. Similar comments can be made for Fig. 2. Litter and fine wood debris (FWD) have no gradient with elevation (Fig. 2a and b) whereas coarse woody debris (CWD) if has a gradient, it is not linear. In Fig. 2b, CWD shows only scattered points. Figure 4a shows no relationship between stand age and elevation: points are too scattered. Even Fig. 2b shows no real relationships between carbon storage of the different components and stand age.

Many other comments could be made on the manuscript but I would point out only the main weaknesses.

---

## Author Comment (AC1) · 24 Sep 2018

The final response to the comments of Referee #1 and Referee #2 was unploaded as a supplement.

Please also note the supplement to this comment:
https://www.biogeosciences-discuss.net/bg-2018-242/bg-2018-242-AC1-supplement.pdf
* * *

---

## Author Comment (AC2) · 24 Sep 2018

Authors' response to reviewers' comments on the manuscript bg-2018-242 "Changes in carbon
stocks of *Fagus* forest ecosystems along an altitudinal gradient on Mt. Fanjingshan in
Southwest China" by Qiong Cai et al.
**To the editor:**
Dear Dr. Frank Hagedorn,
Thank you very much for your treatment of the manuscript and the insightful suggestions from
the two reviewers. These comments were replied focusing on several primary points: (1)
explaining the reasonability of the experimental design and statistical analyses, (2) discussing
the possible effects of management or disturbance, (3) proving the reasonability of stand age
estimation, (4) exploring the possible impacts of soil properties and tree density, (5) updating
the allometric equations for shrubs.
We have carefully addressed these comments in the revised manuscript. Please find our point-
to-point responses to these comments as attached at the bottom of this letter.
We are looking forward to receiving your decision.
Best wishes,

Chengjun Ji
Department of Ecology
Peking University, Beijing 100871, China
Tel: +86-10-6276 5578, Fax: +86-10-6275 6560
E-mail: jicj@ pku.edu.cn

**To Anonymous Referee #1:**

**[Comment] General comments**

The paper is overall well written with the exception of some confused references to Tables and Figures. It is within the scope of BG addressing the question of drivers of forest ecosystem C stock changes. The C stocks estimates in living biomass, dead wood, litter and soil along the transect were based on thorough and comprehensive vegetation and soil sampling and analysis. To place the findings in the context of similar Fagus Ecosystem worldwide, a collection of published data was used. Beyond the estimation of C stocks along the transect, the paper presents no novel methods or insights in mechanisms to explain observed pattern, it can confirm existing knowledge. Given the large range of C storage in different C pools in the examined system and also in the systems used for comparison, the value of the comparison with other studies is limited. In the comparison it may have been interesting to focus and expand on differences and their causes, e.g. regarding the contribution of litter and dead wood. The roles of forest management and use intensity were only moderately addressed, particularly in the comparison with other forests. It may also have been valuable to place the results in the context of the National Forest Resource Inventory database, cf. Fang J, Chen A, Peng C, Zhao S, Ci L (2001) Changes in Forest Biomass Carbon Storage in China Between 1949 and 1998 Science 292:2320-2322 doi:10.1126/science.1058629.

[Reply] Thank you for your insightful comments. Firstly, we are sorry for the confusion caused by the errors in citation the tables and figures. Such mistakes have been avoided in the revised manuscript.

The main purpose of this paper was to provide basic and comprehensive data of the C pools of *Fagus* (beech) forests on Mt. Fanjingshan, a place quite unique and ideal for studies of Chinese beech forests as it has the widest elevational range of Chinese beech forests at a local scale of any region. There have been few reports about the C storage of beech forests in China, compared to other regions (Mund, 2004; Poivesan et al., 2005; Takadi, 1969; Martin and Bailey, 1999; Jenkins et al., 2001). Additionally, the elevation transect provided an excellent environmental gradient to explore the responses of beech forests to varied environmental conditions at a local scale (Körner, 2007).

And we summarized the following three points to reply to your main comments and suggestions.

**1) Comparison with beech forests in other regions**

The comparison of beech forests in different regions was aimed to give a glimpse at the C storage of beech forests on Mt. Fanjingshan at a local scale. To make the comparison more reasonable, we have confined the range of stand age, thus only beech forests within 30–215 year were included [Line 162-163; Table S3]. In the comparison, quantitative analyses of the impacts of management or disturbance were not conducted, considering the limited experimental data and lack of exact documents of disturbance in some sites, despite their significant impacts on C storage of forest ecosystems (Mund, 2004; Mund and Schulze, 2006). Hopefully, it will be discussed in further studies.

**2) The contribution of woody debris and litter**

We have pointed out that the contribution of plant debris in beech forests on Mt. Fanjingshan (< 4%) was comparable to that of forests in China on the whole, while it was relatively lower than that in some temperate forests in other regions of the world (8–47%), possibly because of the differences in stand age and disturbance history (Zhu et al., 2017a). For example, the studies in other countries might include stands that were very old (e.g., Spies and Franklin, 1988) or had suffered catastrophic disturbances (Nalder and Wein, 1999) [Line 264-268].

**3)Impacts of management or human disturbance**

The impacts of management or human disturbance on the elevational patterns of woody debris were further discussed in the Discussion section in the revised manuscript. And we supposed that the elevational patterns of woody debris C storage might be shaped by stand age, disturbance and climate together. [Line 312-323: 'However, it is noteworthy that the C storages of woody debris in several old forests were extremely low (Figure 2c), especially at 1580 m (0.2 Mg C ha$^{-1}$), possibly caused by human disturbance as the plot was not far from a rest platform for the tourists. With this plot excluded (1580 m), the C storage of woody debris was positively related to MAP ($R^2 = 0.66$, $P = 0.01$) and vegetation C storage ($R^2 = 0.68$, $P = 0.01$). Stand age also had a slight positive impact on it despite not statistically significant ($R^2 = 0.39$, $P = 0.1$). The amount of woody debris in the two post-fire young beech forests on Mt. Fanjingshan was quite low, probably because there was little residual woody debris of the previous stands. Besides, the two regenerating stands were young thus had shorter time of accumulation of woody debris. With the increase of elevation, stands tended to be older with more larger trees, resulting in more tree mortality (the self-thinning process) thus increased input of woody debris (Sturtevant et al., 1997), except for the quite low stands (1580 m) possibly disturbed by management activities (collect or removal of woody debris)'].

For the patterns of vegetation and soil, we supposed they were more related to disturbance or management in the past. As in recent decades, beech forests on Mt. Fanjingshan have seldom been disturbed by human management activities such as selection or clear cutting, which were not permitted in the National Natural Reserves. And to some extent, stand age was also an indicator of past human disturbances (Bradford et al., 2008). Besides, we have also discussed the possible impacts of past disturbance on the C storage of soil [Line 346-352: 'The comparatively high soil C storage at the lower two young plots, especially that at 1136 m (229.7 ± 81.3 Mg C ha$^{-1}$) was noteworthy, which was probably related to the previous land use and disturbance of the beech forests. As abovementioned, the lower two beech forests were post-fire secondary forests, which might have accumulated large quantities of C in soil before the dramatic disturbance (Paul et al., 2002; Nave et al., 2010). In addition, plots at lower elevations generally suffered more from human disturbance (Zhang et al., 2009; Alves et al., 2010).'].

**Specific comments / technical corrections**

**[Comment] 1.** Abstract, l.22: Rephrase; the wording is too general. The study presents reliable data on C storage as one among many ecosystem functions but not for understanding structure and function of Chinese beech forests.
**[Reply]** Thank you for your suggestion. We have rephrased the expression as following: 'The present study provides reliable data for understanding the C storage of Chinese beech forests and their possible roles in regional C cycling' [Lines 22-23].

**[Comment] 2.** l.29-30: The summary for policy-makers in IPCC 2013 is not an appropriate reference here. Technically, there are 5 pools since vegetation is separated into above- and below-ground parts. Consider revising the sentence and citing IPCC (2006) 2006 IPCC Guidelines for National Greenhouse Gas Inventories Volume 4 Agriculture, Forestry and Other Land Use. Available at: http://www.ipccnggip.iges.or.jp/public/2006gl/vol4.html.

**[Reply]** Thanks. We have revised it according to your suggestion [Lines 29-30: 'There are different components of C stock in a natural forest ecosystem: vegetation (including aboveground and belowground biomass), woody debris, litter, and soil (IPCC, 2006)'].

**[Comment]** l.33: Delete 'even'.

**[Reply]** We have deleted it.

**[Comment] 3.** l.45 and throughout the manuscript: altitudinal, altitude etc. Consider replacing with elevation, which is more appropriate term in this context; cf. McVicar T, Körner C (2013) On the use of elevation, altitude, and height in the ecological and climatological literature Oecologia 171:335-337 doi:10.1007/s00442-012-2416-7

**[Reply]** Thank you for your suggestion. Throughout the revised manuscript, 'altitude' and 'altitudinal' have been replaced by 'elevation' or 'elevational' (McVicar and Körner, 2013).

**[Comment] 4.** l.50-51: grammar – 'there has been …pattern' or 'there have been…patterns'.

**[Reply]** Thanks. We have revised it as 'there have been no consistent elevational patterns' [Line 51].

**[Comment] 5.** l.56: Consider revising: 'less C accumulation in total' since in relative terms younger stands tend to accumulate more carbon'. Also, biomass accumulation is likely to peak in a mature forest before declining again; cf. Pregitzer and Euskirchen 2004 cited in the manuscript.

**[Reply]** Thanks. We have revised it as you suggested [Line 57].

**[Comment] 6.** l.64: Does 'unneglectable' exist, maybe revise to 'negligible' or 'insignificant'

**[Reply]** We have revised it to 'significant' [Line 64].

**[Comment] 7.** l.90: Is 'consecutive' appropriate; consider 'continuous'.

**[Reply]** Thanks. We have changed it to 'continuous' [Line 73, 90].

**[Comment] 8.** l.149: It would have been interesting to give some indication on the variability of stand age to demonstrate how appropriate this estimate is for the primary and possibly uneven-aged forests. Or possibly a description of the presence/absence of different age cohorts.

**[Reply]** Thanks for your suggestion. Uneven-aged, mixed forests have been reported to account for more than 90% of the forest area worldwide (Dixon, 1994; Bradford et al., 2008). To estimate the stand age of such kinds of forests, two ways are commonly used: the observed tree age or time since disturbance (Bradford et al., 2008). Due to lack of history documents, it is usually difficult to know exactly the disturbance history of the stands especially for old forests.

For the former method, there are also different selections: the maximum age of the largest trees
(Bradford et al., 2008), the average age of the largest 3-5 trees (Bradford et al., 2008), or the
age of the fifth largest tree (Bruelheide et al., 2011; Zhu et al., 2017a). Such selections are all
based on the assumption that DBH accumulated with time.
The DBH-age relationship of the *Fagus* trees on Mt. Fanjingshan was plotted based on the
sampled tree cores. On local scale, trees with larger DBH tended to be older ($R^2 = 0.77$, $P <$
$0.001$; Figure 5a). Positive patterns also existed in most of the plots, and some insignificant
relationship (plots at 1735m) might be due to the limited numbers of sampled trees (Table R1).
In the present study, the age of the fifth largest beech tree was used to stand for the stand age,
and it has been proved to be an excellent indicator of the successional stages of the forests
(Bruelheide et al., 2011).

[Figure]

**Figure 5.** The relationships between (a) diameter at breast height (DBH) and age of the beech
trees, (b) large tree (DBH $\geqslant$ 30 cm) density and stand age.

**Table R1** Summary of relationships between DBH and age of beech trees in each plot on Mt.
Fanjingshan.

| Elevation (m) | $R^2$ | $F$ value | $P$ value | N |
|---|---|---|---|---|
| 1095 | 0.66 | 15.73 | 0.004 | 10 |
| 1136 | 0.52 | 8.659 | 0.019 | 10 |
| 1221 | 0.38 | 4.287 | 0.077 | 10 |
| 1401 | 0.84 | 36.27 | 0.001 | 10 |
| 1500 | 0.98 | 217.3 | 0.000 | 7 |
| 1580 | 0.55 | 4.842 | 0.093 | 6 |
| 1735 | 0.52 | 4.398 | 0.104 | 6 |
| 1843 | 0.91 | 59.97 | 0.000 | 8 |
| 1930 | 0.70 | 14.26 | 0.009 | 8 |

However, it has to be acknowledged that the estimation of stand age might have some
errors and possibly be underestimated. Because only beech trees were sampled considering
their dominant state. Besides, not all largest beech trees were cored due to some sampling
difficulties. Nevertheless, the stand age as estimated in the present study was significantly
positively correlated to the DBH of the fifth largest tree in each plot (linear regression: $R^2 =$
$0.76$, $P = 0.002$).

**[Comment] 9.** l.160: The reference to Tab.2 is not clear in this context as it shows C cocks in the soil, which are not referred to in the preceding sentence, or in this chapter, which is about vegetation.

**[Reply]** We are sorry for the mistakes. In the revised manuscript, such confusions have been avoided.

**[Comment] 10.** l.162, 166, Fig.1: $R^2$ is presented which only tells how well the model is fitting the actual data. In addition, it would be valuable to state whether the coefficients are different from 0. This applies also to the results presented in sections 3.2 and 3.3.

**[Reply]** Thanks. The slopes of linear models are significantly different from 0 in our studies because $P$ value of the slope is the same as that of the model in a general linear model (Figure 2, 3 and 4 in the revised manuscript).

**[Comment] 11.** l.181: Consider inserting 'total soil C storage'.

**[Reply]** Thanks. We have revised it [Lines 192-193].

**[Comment] 12.** l.194, 195, 198, 201: The information is not in Tab. 2, possibly you are referring to Fig. 4?

**[Reply]** Thanks. We have revised the mistakes in the table and figure citation.

**[Comment] 13.** l.204: Is worldwide really appropriate? The comparison of beech forests is based on European, Japanese and Chinese data. The American data are not mentioned in this section.

**[Reply]** Thanks for your comments. We have revised it as 'in other regions'. Actually, beech forests in America were also included, but the available data was quite few. Therefore, the data of America were not included in the table but they were listed in the notes after the table (Table 4 in the revised manuscript).

**[Comment] 14.** l. 208: Figure 3 was not previously introduced and is probably incorrectly cited here as Fig. 5 contains the relevant information.

**[Reply]** We are sorry for the confusion. This figure displayed the distribution of the beech forests on Mt. Fanjingshan and those used for comparison in other regions, thus we thought it make sense and still kept it in the revised manuscript (Figure 1).

**[Comment] 15.** l.222-223: The discussion could be extended to the effect of different management practices and intensities of use. Many beech forests in Europe have been heavily used in the past and show legacies but are now often under some form of protection; cf. Mund M, Schulze E-D (2006) Impacts of forest management on the carbon budget of European beech (Fagus sylvatica) forests Allgem Forst- und Jagdzeitung 177:47-62 and also Mund 2004 cited in the manuscript.

**[Reply]** Thank you for your suggestion. As has been stated above, in the revised manuscript, the possible effects of forest management (removal of woody debris) on the age patterns of woody debris was discussed (4.3 of the Discussion section) [Line 305-327]. For the patterns of
vegetation and soil, we supposed they were more related to disturbance or management in the
past. Please refer to the last two paragraphs in the reply to the special comments.
**[Comment] 16.** l.234-236: Consider moving this to the results section.
**[Reply]** Thanks for your suggestion. We have moved the analysis to the Results section [Line
215-228].
**[Comment] 17.** l.246-249: Please clarify this sentence 'At the same time…lower output'. The
study did not measure decomposition rate. What is the meaning of output, C emissions? This
was not measured. Possibly rephrase to indicate that this is a hypothesis as, for example, on
l.250-251.
**[Reply]** Thanks. The output here means the loss of plant debris, caused by decomposition and
natural or human disturbances. We have rephrased the sentences in the revised manuscript [Line
305-309, 329-336].
**[Comment] 18.** l.247-248: Please clarify this sentence 'Herein, …Input of plant debris'. It is
not clear how increased C storage can result in in increased input of plant debris. Turnover of
tree or shrub was not measured, and if it was the objective to discuss this aspect, a reference to
literature such as Shaozhong Wang, Zhengquan Wang, Jiacun Gu 2017. Variation patterns of
fine root biomass, production and turnover in Chinese forests. Journal of Forestry Research, 28:
1185-1194 may be appropriate.
**[Reply]** Thanks. We have rephrased the sentences. Although biomass has been found to be
positively correlated with the C storage of woody debris both on national (Zhu et al., 2017a)
and local scales (Zhu et al., 2017b), one of the direct input of coarse woody debris is from tree
mortality (Spies and Franklin, 1988). Actually, the amount of woody debris is generally
determined by the timing of inputs, decomposition rate and the amount removed by natural or
human disturbance (Harmon et al., 1986; Spies and Franklin, 1988). The inputs may be
inherited from the previous stand after catastrophic disturbances, or recruit from tree mortality
during the succession course (Spies and Franklin, 1988; Bond-Lamberty et al., 2002). And the
latter tends to increase before the forest senesces (Sturtevant et al., 1997). Herein, with the
increase of elevation, stands tended to be older with more large trees, resulting in more tree
mortality (the self-thinning process) thus increased input of woody debris (Sturtevant et al.,
1997) [Line 305-336].
The case may be somewhat different for litter. The amount of litter is mainly determined
by the input of litter fall and the output through decomposition or disturbances (e.g., removal).
Litter accumulation was found to be strongly related to canopy cover (Hall et al., 2006), while
its relationships with biomass or stand age were not consistent in previous studies (Peichl and
Arain, 2006; Zhu et al., 2017a, b). And the decomposition rate is generally faster than woody
debris. Herein, the canopy cover had no obvious elevational patterns. Previous studies
suggested that the relative faster decay rate of litter might enable it to reach a balance between
the input and output more quickly (Zhu et al. 2017b). However, for more exact explanation, the
data of litter fall and decomposition rate are required [Line 329-336].

**[Comment] 19.** l.253: Fig. 2 does not include information on soil.
**[Reply]** We are sorry for the mistake and have revised it in the updated manuscript.
**[Comment] 20.** l.264 & 182: The fact that the two secondary forests were disturbed by fire
may explain the comparatively high soil C as there may be fire-derived carbon.
**[Reply]** We agree with you, and it has been stated in the manuscript as 'the lower two beech
forests were post-fire secondary forests, which might have accumulated large quantities of C in
soil before the dramatic disturbance (Paul et al., 2002; Nave et al., 2010). In addition, plots at
lower elevations generally suffered more from human disturbance (Zhang et al., 2009; Alves et
al., 2010)' [Line 349-352].
**[Comment] 21.** l.265-267: The link between the first and second subclause is not clear. The
reference Pregitzer and Euskirchen is not correct here, as they do not demonstrate a relationship
between disturbance intensity and elevation.
**[Reply]** Thanks, we have rephrased the sentences [Line 352-355].
**[Comment] 22.** l.267-268: 'therefore' is not appropriate as this it is a hypothesis.
**[Reply]** Thanks. We have deleted it.
**[Comment] 23.** l.278-285: A further limitation is the uncertainty related to the application of
allometric equations to estimate tree biomass. Standard deviations are presented for soil C in
Tab. 2.
**[Reply]** Thank you for your insightful comments. We have addressed the possible limitation of
using allometric equations in the Discussion section [Line 370-373: 'Furthermore, the C storage
of trees and shrubs were estimated using allocation equations as destructive sampling was
forbidden. This could have resulted in some estimation errors despite careful selection, because
the equations might be closely related to regions, forest types and species (Lima et al., 2012).'].
**[Comment] 24.** l.297-298: It is not clear how the study contributes to the understanding of
structure and function of beech forests in China. Rephrase to something like 'C storage and
distribution among pools'.
**[Reply]** Thanks. We have revised the expression as 'understanding the C storage of Chinese
beech forests and their possible roles in regional C cycling' [Line 387].
**[Comment] 25.** Table 1: What is the explanation of the comparatively low age of 88 years of
stand FJ4 relative to the other primary forests?
**[Reply]** The estimation of stand age might have some bias and possibly underestimation due to
the difficulties in sampling the largest trees sometimes. However, the estimated stand age was
positively related to the DBH of the 5th largest tree, which has been used to stand for the
succession stage (Bruelheide et al., 2011). For more explanation, please refer to the response to
Comment 8.
**[Comment] 26.** Table 2: Please indicate different meanings of the letters a and b, which are
used to indicate significant differences.

**[Reply]** In the table (Table 3 in the revised manuscript), different letters indicate significant
difference among soil layers in each column ($P < 0.05$), as is generally adopted in the statistical
analysis. Here, values marked with 'a' is significantly larger than that marked with 'b', that is
to say, alphabetical letters indicate the values from large to small.
**[Comment] 27.** Tab. 3: The data for American beech forest are missing.
**[Reply]** The data for American beech forests are listed in the Note of Table 3 (Table 4 in the
revised manuscript).
**[Comment] 28.** Figure 3: It appears that this figure is never referred to in the text; the reference
to fig 3 on l.208 appears to refer to Fig. 5.
**[Reply]** Thanks. As we have addressed (reply to Comment 14), this figure (changed to Figure
1 in the revised manuscript) displayed the distribution of the beech forests on Mt. Fanjingshan
and those used for comparison in other regions. In the revised manuscript, it is cited in Line
163.
**[Comment] 29.** Fig.4: Consider enlarging the figure or placing legend and coefficients
differently.
**[Reply]** Thanks for your suggestion. The figure has been modified to make it more clear (Figure
3 in the revised manuscript).
**[Comment] 30.** Table S2: What are the reasons and the effect of modifying the equations? How
reasonable is it to use root: shoot ratios of trees for shrubs? This may not be appropriate, cf.
Mooney HA (1972) The Carbon Balance of Plants Annual Review of Ecology and Systematics
3:315-346, and should be discussed as limitation and source of error.
**[Reply]** Thank you for your valuable comments. In the former manuscript, the equations were
modified based on the shrub samples we collected in the plots. However, the number of samples
were limited thus it might result in some bias. Besides, it is true that the allocation strategies of
different life forms might be distinct (Mooney, 1972). Therefore, we have recalculated the
biomass of shrubs using new allocation equations which include both aboveground biomass
(AGB) and belowground biomass (BGB) (Dong et al., 2002; Zhao, 2012; Tu et al., 2015; Xie
et al., in press) (Table S2). Accordingly, the values of the C storage of vegetation and total
ecosystem have also been updated in the revised manuscript. And the elevational trend of the
shrub C storage was the same despite the absolute values varied a little. The new equations are
listed in Table S2 as follows.
In the revised manuscript, the limitations of using allocation equations have also been
addressed (Please see reply to Comment 23).

**Table S2.** Equations for calculating aboveground biomass (AGB, kg) and belowground
biomass (BGB, kg) of dominant shrub species used in this study. D, diameter at shoot base (cm);
H, height of a shrub (m); A, crown area of a shrub ($m^2$); V, projected volume of a shrub ($m^3$),
V= AH.

| Species | Biomass equation | Reference |
|---|---|---|
| *Ardisia* | $AGB=0.004+0.137V+0.223V^2$ | Zhao, 2012 |
| | $BGB=0.001+0.122V+0.038V^2$ | |
| *Castanopsis, Lithocarpus* | $AGB=0.067(D^2H)^{0.7039}$ | Xie et al., in press |
| | $BGB=0.3446AGB^{0.7871}$ | |
| *Cyclobalanopsis, Fagus* | $AGB=0.0603+0.0274\ D^2H$ | Xie et al., in press |
| | $BGB=0.3866AGB^{0.753}$ | |
| Ericaceae | $AGB=0.0494(D^2H)^{0.7627}$ | Xie et al., in press |
| | $BGB=0.5483AGB^{0.8124}$ | |
| Rosaceae | $AGB=0.0602(D^2H)^{0.5989}$ | Xie et al., in press |
| | $BGB=0.1879AGB^{0.7329}$ | |
| *Rubus* | $AGB=0.0362(D^2H)^{0.7555}$ | Xie et al., in press |
| | $BGB=0.1096ABG^{0.672}$ | |
| Theaceae | $AGB=0.0613(D^2H)^{0.7102}$ | Xie et al., in press |
| | $BGB=0.4014AGB^{0.47451}$ | |
| *Yushania brevipaniculata* | $AGB=(132.92D1.36+32.7768D-8.1026+6.6254\ D^2H)/1000$ | Dong et al., 2002 |
| | $BGB=(10.5903(D^2H)^{0.5207}+57.2177(D^2H)^{0.2676}+21.0077(D^2H)^{0.4024})/1000$ | |
| Liana | $AGB=0.0581(D^2H)^{0.9384}$ | Xie et al., in press |
| | $BGB=0.0292(D^2H)^{0.7569}$ | |
| Other species | $AGB=(35.4+0.0419(DH)+0.00203(DH)^2-0.00000108(DH)^3-BGB)/1000$ $-BGB$ | Tu et al., 2015 |
| | $BGB=(9.64+0.0703(DH)+0.000546(DH)^2-0.000000296(DH)^3)/1000$ | |

**To Anonymous Referee #2**

[Comment] The manuscript could a contribution of interest for Biogeosciences and in principle within its specific scope but it is not suitable for publication in this form. The Authors have made a great effort in collecting many data, but the experimental design is not appropriate to the proposed objectives. In addition, statistical analysis of data is poor and misused because the statistical results do not confirm what the Authors reported as main findings. The Authors have completely neglected spatial variability in both soil and forest cover. Organic carbon content in soils is strongly dependent on the soil properties and particularly, on soil texture. The Authors should provide at least the main soil properties of the nine areas to show their homogeneity. Such an implicit assumption of homogeneity cannot hold with no information on soil properties. The nine stands are not comparable because they have different ages (ranging between 44 and 185 years), density (ranging between 1483 and 2350 stems/ha). How do the Authors think possible to evaluate the key driving factors of altitude gradient in vegetation carbon storage? To separate the elevation effect from other attributes, it is requested to have homogeneous stands in which the only variable factor is elevation.

[Reply] Thank you for your insightful comments. We have tried our best to improve the analyses and we will address our reply as follows:

**1) About the experimental design**

As we have stated in the manuscript, along the elevation, there forms a complex environmental gradient, homogeneous stands are hard to find in natural state (Körner, 2007) [Line 43-45]. Such dilemma has also been faced in many previous studies exploring the elevational patterns (e.g., Zhu et al., 2010; Alves et al., 2010). Actually, ecological patterns and processes in the field are generally affected by complex biotic and abiotic factors. Researchers usually tried to find out the possible driving factors by conducting suitable statistical analyses, such as linear or nonlinear regression (Zhu et al., 2010), stepwise multiple regression (Zhang et al., 2009), generalized linear model (GLM) (Yang et al., 2008), partial GLM (Ma et al., 2018), or even more complex methods like structural equation modeling (Xu et al., 2018). And usually one or several of the abovementioned methods were adopted.

**2) Improvements of the statistical analyses to explore possible driving factors**

The limited number of data prevented us from conducting more reasonable statistical analyses, however, we still made some efforts to explore the factors that have relatively stronger impacts on the elevational patterns of different C pools. Firstly, the effects of individual factor on different C components were explored using linear regressions. Then stepwise multiple regressions were further conducted to determine the relative strength and direction of the effects of multiple factors for the C storage of vegetation and woody debris (Paoli and Curran, 2007; Zhang et al., 2009). Prior to the stepwise regression, the variables were all normalized to make the regression coefficients comparable:

$$x' = \frac{x - \min(x)}{\max\ (x) - \min(x)}$$

Then, in the multiple linear model, the factor with relatively larger absolute regression coefficient may have stronger impacts on or be a better predictor of the variation of the C

storage of vegetation or woody debris [Line 214-240].

**3) Soil properties**

Information about the soil properties has been complemented based on laboratory
determination (Table R2), and some further analyses have also been conducted. Herein, soil C
storage was found to positively correlated with soil C concentration ($R^2 = 0.45$, $P < 0.001$), N
concentration ($R^2 = 0.49$, $P < 0.001$) and bulk density ($R^2 = 0.20$, $P = 0.02$), while not related
to moisture and C: N ratio ($P > 0.05$). And these soil properties showed weak impacts on the
vegetation C storage ($P > 0.05$).

Unfortunately, we did not have the data of soil texture (such as silt content and clay
content), but the soil types were almost the same for the plots (Editorial Board of the Scientific
Survey of the Fanjingshan Mountain Preserve Guizhou Province, China, 1986). Besides,
previous studies have shown that impacts of soil texture on productivity (thus biomass) might
be related to moisture and nutrient availability (de Castilho et al., 2006). Thus, the possible
effects of soil texture and other properties (e.g., pH) still remain to be explored in further studies
[Line 352-355].

**Table R2** Soil properties of the beech forests on Mt. Fanjingshan

| Altitude (m) | Bulk density (g cm$^{-3}$) | Soil moisture | C concentration | N concentration | C: N ratio |
|---|---|---|---|---|---|
| 1095 | 0.48±0.1 | 0.44±0.03 | 6.69±1.64 | 0.52±0.12 | 12.77±0.67 |
| 1136 | 0.78±0.22 | 0.39±0.09 | 7.86±1.19 | 0.54±0.12 | 14.7±1.13 |
| 1221 | 0.37±0.1 | 0.55±0.07 | 7.62±4.43 | 0.52±0.26 | 14.13±1.62 |
| 1401 | 0.37±0.06 | 0.47±0.05 | 6.94±1.26 | 0.47±0.07 | 14.57±0.42 |
| 1500 | 0.6±0.24 | 0.39±0.08 | 4.37±1.48 | 0.35±0.1 | 12.29±0.72 |
| 1580 | 0.55±0.14 | 0.46±0.02 | 8.42±2.76 | 0.55±0.16 | 15.26±1 |
| 1735 | 0.39±0.17 | 0.57±0.11 | 4.16±0.73 | 0.37±0.07 | 11.26±0.13 |
| 1843 | 0.61±0.13 | 0.49±0.05 | 3.92±0.17 | 0.36±0.01 | 10.9±0.15 |
| 1930 | 0.52±0.12 | 0.51±0.05 | 6.64±0.82 | 0.44±0.04 | 15.21±0.53 |

**4) Forest coverage and stem density**

Forest coverage was roughly estimated in each plot, and it showed no significant elevational
patterns (Table 1). Stem density also had no obvious elevational patterns despite a large
variation (1483 and 2350 stems ha$^{-1}$) (Table 1). Both of them had little effects on the variation
of the four C components, despite their significant impacts in previous studies (e.g., Hall et al.,
2006). It has to be acknowledged that the estimation of coverage might have some errors, thus
its impacts on the storage of different C pools still need further research.

And for stem density, stems with a DBH ⩾ 3 cm were regarded as trees. In the plots,
small trees usually accounted for a large proportion of the stems while their contribution to
biomass were relatively small. Thus, we further discussed the contribution of large trees (DBH
⩾ 30 cm) to vegetation C storage (DeWalt and Chave, 2004; Xu et al., 2015). Large tree
density showed positive relationships with elevation ($R^2 = 0.67$, $P < 0.01$; Table 2), and their
contribution to biomass also increased at higher beech forests (Table 2) [Line 177-183]. Large
tree density tended to increase in older beech forests ($R^2 = 0.78$, $P = 0.002$; Figure 5b) and
contributed greatly to the increase of vegetation C storage. More detailed discussions about this were added in the Discussion section [Line 282-288].

**Table 2.** Density of large trees (No. ha$^{-1}$) (DBH ≥ 30cm) and their contributions to tree and vegetation carbon (C) storage in beech forests on Mt. Fanjingshan. TBA, total basal area.

| Elevation (m) | Density (No. ha$^{-1}$) | Percentage of stems | Percentage of TBA | Percentage of tree C | Percentage of vegetation C |
|---|---|---|---|---|---|
| 1095 | 33 | 1.4% | 15.3% | 13.6% | 13.4% |
| 1136 | 83 | 3.9% | 19.7% | 15.7% | 15.6% |
| 1221 | 150 | 10.1% | 47.1% | 46.9% | 46.6% |
| 1401 | 117 | 6.4% | 48.8% | 48.6% | 48.3% |
| 1500 | 283 | 14.7% | 75.4% | 79.0% | 77.1% |
| 1580 | 300 | 18.0% | 84.6% | 87.4% | 86.0% |
| 1735 | 283 | 12.1% | 63.2% | 62.7% | 61.0% |
| 1843 | 167 | 8.2% | 65.5% | 66.8% | 65.5% |
| 1930 | 250 | 15.5% | 80.2% | 84.8% | 83.9% |

**[Comment]** Results are not supported by statistical analysis. The changes in vegetation carbon storage along the elevation gradient makes no sense (Fig. 1). A simple visual inspection of Fig.

1a for trees data, shows a regression line through two cluster of points and reporting significant coefficient of determination has no statistical meaning. The same occurred for the aboveground vegetation (Fig. 1b). Shrub and herb have no gradient (Fig, 1a): the regression line is almost horizontal. Similar comments can be made for Fig. 2. Litter and fine wood debris (FWD) have no gradient with elevation (Fig. 2a and b) whereas coarse woody debris (CWD) if has a gradient, it is not linear. In Fig. 2b, CWD shows only scattered points. Figure 4a shows no relationship between stand age and elevation: points are too scattered. Even Fig. 2b shows no real relationships between carbon storage of the different components and stand age. Many other comments could be made on the manuscript but I would point out only the main weaknesses.

**[Reply]** Thank you for your insightful comments. We will respond to your comments based on the following points.

**1)  Why we focused on the elevational patterns**

As abovementioned, the primary purpose of our study was to provide basic data of the beech forests on Mt. Fanjingshan, as there have been few reports about the C storage of beech forests in China, compared to other regions in the Northern Hemisphere (Mund, 2004; Poivesan et al.,

2005; Takadi, 1969; Martin and Bailey, 1999; Jenkins et al., 2001). Mt. Fanjingshan is a place quite unique and ideal for studies of Chinese beech forests as it has the widest elevational range of Chinese beech forests at a local scale of any region. Such an elevation transect provides an excellent environmental gradient to explore how beech forests respond to varied environmental conditions at a local scale (Körner, 2007).

**2)  The reasonability of the statistical analyses**

Owing to the limited quantities of experimental data, the points might seem clustered or scattered, and the statistical analyses were relatively simple. However, we have tested the normality (Shapiro-Wilk test) of the experimental data and the results showed that most of the
variables, excluding the C storage of soil (Figure R1), obeyed a normal distribution (Table R3).
Thus, we supposed the linear regression analyses were reasonable. And the judgements of the
elevational trends or the relationships between different variables were all based on the $P$ value
of the statistical analyses. For example, the stand age tended to increase with increasing
elevation ($R^2 = 0.56$, $P = 0.02$; Figure 3a in the revised manuscript).

**Table R3** Normality test (Shapiro-Wilk) of the variables

| Components | W Statistic | df | $P$ value |
|---|---|---|---|
| Vegetation | 0.883 | 9 | 0.170 |
| Tree | 0.889 | 9 | 0.196 |
| Shrub | 0.852 | 9 | 0.079 |
| Herb | 0.853 | 9 | 0.081 |
| AGB | 0.887 | 9 | 0.185 |
| BGB | 0.897 | 9 | 0.234 |
| Litter | 0.966 | 27 | 0.490 |
| Woody Debris | 0.841 | 9 | 0.060 |
| **Soil** | **0.664** | **27** | **0.000** |
| Ecosystem | 0.918 | 9 | 0.372 |
| Stand age | 0.901 | 9 | 0.258 |

Note: the data are supposed to obey normal distribution with $P > 0.05$

[Figure]

**Figure R1** Frequency distributions of soil carbon storage

**References**

Alves, L. F., Vieira, S. A., Scaranello, M. A., Camargo, P. B., Santos, F. A., Joly, C. A., and Martinelli, L. A.: Forest structure and live aboveground biomass variation along an elevational gradient of tropical Atlantic moist forest (Brazil), Forest Ecol. Manag, 260, 679–691, 2010.

Bond-Lamberty, B., Wang, C., and Gower, S. T.: Annual carbon flux from woody debris for a boreal black spruce fire chronosequence, J. Geophys. Res-Atmos., 107, WFX-1, 2002.

Bradford, J. B., Birdsey, R. A., Joyce, L. A., and Ryan, M. G.: Tree age, disturbance history, and carbon stocks and fluxes in subalpine Rocky Mountain forests, Glob. Chang. Biol., *14*, 2882–2897, 2008.

Bruelheide, H., Böhnke, M., Both, S., Fang, T., Assmann, T., Baruffol, M., ... and Bernhard, S.: Community assembly during secondary forest succession in a Chinese subtropical forest, Ecol. Monogr, 81, 25–41, 2011.

de Castilho, C. V., Magnusson, W. E., de Araújo, R. N. O., Luizao, R. C., Luizao, F. J., Lima, A. P., and Higuchi, N.: Variation in aboveground tree live biomass in a central Amazonian Forest: Effects of soil and topography, Forest Ecol. Manag., 234, 85–96, 2006.

DeWalt, S. J., and Chave, J.: Structure and biomass of four lowland Neotropical forests, Biotropica, 36, 7–19, 2004.

Dixon, R. K., Brown, S., Houghton, R. A., Solomon, A. M., Trexler, M. C., and Wisniewski, J.: Carbon pools and flux of global forest ecosystems, Science, 263, 185–190, 1994.

Dong, W. Y., Huang, B. L., Xie, Z. X., Xie, Z. H., and Liu, H. Y. Studies on the structure and dynamics of biomass of *Qiongzhuea tumidinoda* clone Population, Forest Research, 15, 416–420, 2002[In Chinese with English abstract].

Editorial Board of the Scientific Survey of the Fanjingshanshan Mountain Preserve, Guizhou Province, China: Scientific Survey of the Fanjingshanshan Mountain Preserve Guizhou Province, China, China Environmental Science Press, Beijing, China, 1986 [In Chinese].

Hall, S. A., Burke, I. C., and Hobbs, N. T.: Litter and dead wood dynamics in ponderosa pine forests along a 160-year chronosequence, Ecol. Appl., 16, 2344–2355, 2006.

Intergovernmental Panel on Climate Change (IPCC): 2006 IPCC Guidelines for National Greenhouse Gas Inventories, vol. 4, Agriculture, Forestry and Other Land Use, edited by: Eggleston, H. S., Buendia, L., Miwa, K., Ngara, T., and Tanabe, K., Inst. for Global Environ. Strategies, Hayama, Japan, 2006.

Jenkins, J. C., Birdsey, R. A., and Pan, Y.: Biomass and NPP estimation for the Mid-Atlantic region (USA) using plot-level forest inventory data, Ecol. Appl., 11, 1174–1193, 2001.

Körner, C.: The use of 'elevation' in ecological research, Trends Ecol. Evol., 22, 569–74, 2007.

Ma, S. H., He, F., Tian, D., Zou, D. T., Yan, Z. B., Yang, Y. L., Zhou, T. C., Huang K. Y., Shen, H. H., and Fang, J. Y.: Variations and determinants of carbon content in plants: a global synthesis, Biogeosciences, 15, 693, 2018.

Martin, C. W., and Bailey, A. S.: Twenty years of change in a northern hardwood forest, Forest Ecol. Manag., 123, 253–260, 1999.

McVicar, T. R. and Körner, C.: On the use of elevation, altitude, and height in the ecological and climatological literature, Oecologia, 171, 335–337, 2013.

Mooney, H. A.: The carbon balance of plants, Annu. Rev. Ecol. Syst., 3, 315–346, 1972.

Mund, M.: Carbon pools of European beech forests (*Fagus sylvatica*) under different silvicultural management, Ph.D. thesis, The University of Göttingen, Germany, 256 pp., 2004.

Nalder, I. A. and Wein, R. W.: Long-term forest floor carbon dynamics after fire in upland boreal forests of western Canada, Glob. Biogeochem. Cy., 13, 951–968, 1999.

Nave, L. E., Vance, E. D., Swanston, C. W., and Curtis, P. S.: Harvest impacts on soil carbon storage in
temperate forests, Forest Ecol. Manag., 259, 857–866, 2010.

Paoli, G. D. and Curran, L. M.: Soil nutrients limit fine litter production and tree growth in mature
lowland forest of southwestern Borneo, Ecosystems, 10, 503–518, 2007.

Paul, K. I., Polglase, P. J., Nyakuengama, J. G., Khanna, P. K.: Change in soil carbon following
afforestation, Forest Ecol. Manag., 168, 241–257, 2002.

Peichl, M. and Arain, M. A.: Above-and belowground ecosystem biomass and carbon pools in an age-
sequence of temperate pine plantation forests, Agr. Forest Meteorol., 140, 51–63, 2006.

Piovesan, G., Di Filippo, A., Alessandrini, A. E. A., Biondi, F., and Schirone, B.: Structure, dynamics
and dendroecology of an old-growth *Fagus* forest in the Apennines, J. Veg. Sci., 16, 13–28, 2005.

Spies, T. A., Franklin, J. F., and Thomas, T. B.: Coarse woody debris in Douglas-fir forests of western
Oregon and Washington, Ecology, 69, 1689–1702, 1988.

Sturtevant, B. R., Bissonette, J. A., Long, J. N., and Roberts, D. W.: Coarse woody debris as a function
of age, stand structure, and disturbance in boreal Newfoundland, Ecol. Appl., 7, 702–712, 1997.

Tadaki, Y., Hatiya, K., and Tochiaki, K.: Studies on the Production Structure of Forest (XV), J. Jpn. For.
Soc., 51, 331–339, 1969[In Japanese with English abstract].

Tu, H. T., Sun, Y. J., Liu, S. Z., Yi, X. T., and Song, F. Establishment of the models on shrub biomass in
Jiangle forest farm, Fujian, Journal of Northwest Forestry University, 30, 89–93, 2015[In Chinese with
English abstract].

Xie, Z. Q., Wang, Y., Tang, Z. Y., and Xu, W. T. Handbook of biomass models of common shrubs in
China, Science Press, Beijing, China, in press [in Chinese].

Xu, Y., Franklin, S. B., Wang, Q., Shi, Z., Luo, Y., Lu, Z., Zhang, J., Qiao, X., and Jiang, M.: Topographic
and biotic factors determine forest biomass spatial distribution in a subtropical mountain moist forest,
Forest Ecol. Manag., 357, 95–103, 2015.

Xu, L., Shi, Y., Fang, H., Zhou, G., Xu, X., Zhou, Y., Tao, J., Ji, B., and Chen, L.: Vegetation carbon
stocks driven by canopy density and forest age in subtropical forest ecosystems, Sci. Total
Environ., 631, 619–626, 2018.

Yang, Y. H., Fang, J. Y., Tang, Y. H., Ji, C. J., Zheng, C. Y., He, J. S., and Zhu, B.: Storage, patterns and
controls of soil organic carbon in the Tibetan grassland, Glob. Chang. Biol., 14, 1592–1599, 2008.

Zhang, X. P., Wang, M. B., and Liang, X. M.: Quantitative classification and carbon density of the forest
vegetation in Lu¨liang Mountains of China, Plant Ecol., 201, 1–9, 2009.

Zhao, B. Biomass and Value of Ten Shrub Species in Dagang Mountai, Master thesis, Beijing Forestry
University, China, 74pp., 2012 [In Chinese with English abstract].

Zhu, B., Wang, X. P., Fang, J. Y., Piao, S. L., Shen, H. H., Zhao, S. Q., and Peng, C. H.: Elevational
changes in carbon storage of temperate forests on Mt Changbai, northeast China, J. Plant Res., 123,
439–452, 2010.

Zhu, J. X., Hu, H. F., Tao, S. L., Chi, X. L., Li, P., Jiang, L., Ji, C. J., Zhu, J. L., Tang, Z. Y., Pan, Y. D.,
Birdsey, R. A., He, X. H., and Fang, J. Y.: Carbon stocks and changes of dead organic matter in China's
forests, Nat. Commun., 8, 1–10, 2017a.

Zhu, J. X., Zhou, X. L., Fang, W. J., Xiong, X. Y., Zhu, B., Ji, C., and Fang, J. Y.: Plant debris and its
contribution to ecosystem carbon storage in successional *Larix gmelinii* forests in northeastern China,
Forests, 8, 191, 2017b.